# SOFT Q-LEARNING WITH MUTUAL-INFORMATION REGULARIZATION

**Jordi Grau-Moya, Felix Leibfried and Peter Vrancx**
PROWLER.io
Cambridge, United Kingdom
`{jordi}@prowler.io`

## ABSTRACT

We propose a reinforcement learning (RL) algorithm that uses mutual-information regularization to optimize a prior action distribution for better performance and exploration. Entropy-based regularization has previously been shown to improve both exploration and robustness in challenging sequential decision-making tasks. It does so by encouraging policies to put probability mass on all actions. However, entropy regularization might be undesirable when actions have significantly different importance. In this paper, we propose a theoretically motivated framework that dynamically weights the importance of actions by using the mutual-information. In particular, we express the RL problem as an inference problem where the prior probability distribution over actions is subject to optimization. We show that the prior optimization introduces a mutual-information regularizer in the RL objective. This regularizer encourages the policy to be close to a non-uniform distribution that assigns higher probability mass to more important actions. We empirically demonstrate that our method significantly improves over entropy regularization methods and unregularized methods.

## 1 INTRODUCTION

Reinforcement Learning (RL) (Sutton & Barto, 1998) is a framework for solving sequential decision-making problems under uncertainty. Contemporary state-of-the-art RL methods often use an objective that includes an entropy regularization term (Haarnoja et al., 2018c; 2017; Teh et al., 2017). Entropy regularized RL has been shown to capture multi-modal behaviour, as well as exhibiting superior exploration (Haarnoja et al., 2017). Additionally, the learned policies are more robust, as the entropy bonus accounts for future action stochasticity (Grau-Moya et al., 2016) and reduces value overestimation (Fox et al., 2016).

While encouraging high-entropy policies can provide several benefits, it is possible to devise examples where entropy regularization actually impedes exploration. Since high-entropy policies tend to spread the probability mass across all actions equally, they can perform poorly when the RL problem contains actions that are rarely useful. In this paper, we propose to overcome the previous limitation by designing a reinforcement learning algorithm that dynamically adjusts the importance of actions while learning. We motivate our algorithm by phrasing RL as an inference problem with an adaptive prior action distribution. Where previous work assumes a uniform prior distribution over the actions (Rawlik et al., 2012; Levine, 2018), we generalize the formulation by optimizing the prior. We show that this optimization process leads to an RL objective function with a regularizer based on the mutual information between states and actions.

Additionally, we develop a novel algorithm that uses such mutual-information regularization to obtain an optimal action-prior for better performance and exploration in high-dimensional state spaces. This novel regularizer for RL encourages policies to be close to the marginal distribution over actions. This results in assigning higher probability to actions frequently used by the optimal policy, while actions that are used infrequently have lower probability under the prior. We demonstrate significant improvements on 19 Atari games over a deep Q-network (Mnih et al., 2015) (DQN) baseline without any regularization and over soft Q-learning (Schulman et al., 2017; Leibfried et al., 2018) (SQL) that employs standard entropy regularization without prior adaptation.

## 2 BACKGROUND

### 2.1 REINFORCEMENT LEARNING

We consider the standard Markov decision process (MDP) setting. Formally, an MDP is defined as the tuple $\langle \mathcal{S}, \mathcal{A}, P, R, \gamma \rangle$ where $\mathcal{S}$ is the state space, $\mathcal{A}$ the action space, and $P : \mathcal{S} \times \mathcal{A} \times \mathcal{S} \to [0, 1]$ denotes the state transition function. Upon taking action $a_t \in \mathcal{A}$ in state $s_t \in \mathcal{S}$, the agent transitions to $s_{t+1}$ with probability $P(s_{t+1}|s_t, a_t)$. The reward function $\mathcal{R} : \mathcal{S} \times \mathcal{A} \to \mathbb{R}$ quantifies the agent's performance. The goal is to find a policy that maximizes the value function, i.e. $\pi^*(a|s) = \arg\max_\pi V^\pi(s)$, where $V^\pi(s) = \mathbb{E}\left[\sum_{t=0}^T \gamma^t r(s_t, a_t)|s_0 = s\right]$. Here, $\gamma$ is a discount factor ($0 < \gamma < 1$) that allows to account for the future in different ways.

The policy-dependent state transition probabilities are defined as $P_\pi(s'|s) := \sum_a P(s'|a, s)\pi(a|s)$ which can be written in matrix notation as $\boldsymbol{P}_\pi \in \mathbb{R}^{|\mathcal{S}|} \times \mathbb{R}^{|\mathcal{S}|}$ where the rows are indexed by $s$ and the columns by $s'$. This allows us to conveniently define the agent's stationary distribution over states $\mu_\pi(s)$ and actions $\rho_\pi(a)$ as follows:

**Definition 1** (Stationary distribution over states). *The stationary distribution over states (assumed to exist and to be unique) is defined in vector form as $\boldsymbol{\mu}_\pi^\top := \lim_{t \to \infty} \boldsymbol{\nu}_0^\top \boldsymbol{P}_\pi^t$ with $\boldsymbol{\nu}_0$ being an arbitrary vector of probabilities over states at time $t = 0$. The stationary distribution satisfies $\mu_\pi(s') = \sum_s P_\pi(s'|s)\mu_\pi(s)$ and therefore is a fixed point under the state transition probabilities $\boldsymbol{\mu}_\pi^\top = \boldsymbol{\mu}_\pi^\top \boldsymbol{P}_\pi$.*

**Definition 2** (Stationary distribution over actions). *Let $\mu_\pi(s)$ be the stationary distribution over states induced by the policy $\pi$. Then the stationary distribution over actions under the policy $\pi$ is defined as $\rho_\pi(a) := \sum_{s \in \mathcal{S}} \mu_\pi(s)\pi(a|s)$.*

### 2.2 MAXIMUM ENTROPY REINFORCEMENT LEARNING

Maximum entropy reinforcement learning augments the standard RL reward objective with an additional policy entropy term. The optimal value function under entropy regularization (Haarnoja et al., 2017) is defined as:

$$\mathcal{V}^*(s) = \max_\pi \mathbb{E}\left[\sum_{t=0}^\infty \gamma^t \left(r(s_t, a_t) - \frac{1}{\beta}\log\pi(a_t|s_t)\right)\bigg|s_0 = s\right], \tag{1}$$

where $\frac{1}{\beta}$ trades off between reward and entropy maximization, and the expectation operation is over state-action trajectories. The optimal policy that solves (1) can be written in closed form as: $\pi^*(a|s) = \frac{e^{\beta\mathcal{Q}^*(s,a)}}{\sum_{a \in \mathcal{A}} e^{\beta\mathcal{Q}^*(s,a)}}$, where $\mathcal{Q}^*(s,a) := r(s,a) + \sum_{s' \in \mathcal{S}} P(s'|s,a)\mathcal{V}^*(s')$. Note that the above represents a generalization of standard RL settings, where $\beta \to \infty$ corresponds to a standard RL valuation ($\lim_{\beta \to \infty} \mathcal{V}^*(s) = \max_\pi V^\pi(s)$), while for $\beta \to 0$ we recover the valuation under a random uniform policy. For intermediate values of $\beta$, we can trade off between reward maximization and entropy maximization.

Interestingly, one can formulate the maximum entropy RL objective as an inference problem (Levine, 2018) by specifying a prior distribution over trajectories that assumes a fixed uniform distribution over actions. Precisely this assumption is what encourages the policies to maximize entropy. As outlined in the introduction, encouraging policies to be close to a uniform distribution might be undesirable when some actions are simply non-useful or not frequently used.

In Section 3, we show that when relaxing the previous assumption, i.e. allowing for prior optimization, we obtain a novel variational inference formulation of the RL problem that constrains the policy's *mutual-information* between states and actions. We show that such policies must be close to the marginal distribution over actions which automatically assigns high probability mass to overall useful actions and low probability to infrequently used actions.

Before proceeding, however, it is insightful to show how prior optimization bridges the gap between entropy regularization and mutual-information regularization in a non-sequential decision-making scenario that considers one time step only.

## 2.3 MUTUAL-INFORMATION REGULARIZATION FOR ONE-STEP DECISION-MAKING

In a one-step decision-making scenario, entropy regularization assumes the following form

$$\max_\pi \sum_{s,a} p(s)\pi(a|s)\left(r(s,a) - \frac{1}{\beta}\log\pi(a|s)\right),$$

where $p(s)$ is some arbitrary distribution over states and the optimal policy balances expected reward maximization versus expected entropy maximization.

Entropy regularization discourages deviations from a uniform prior policy. In a more general setting, when discouraging deviations from an arbitrary prior $\rho(a)$, a similar objective can be written as

$$\max_\pi \sum_{s,a} p(s)\pi(a|s)\left(r(s,a) - \frac{1}{\beta}\log\frac{\pi(a|s)}{\rho(a)}\right) =$$

$$\max_\pi \sum_{s,a} p(s)\pi(a|s)r(s,a) - \frac{1}{\beta}\sum_s p(s)\mathrm{KL}(\pi(\cdot|s)||\rho(\cdot)), \tag{2}$$

where KL refers to the Kullback-Leiber (KL) divergence. This framework has been proposed before in the literature under the name information-theory for decision-making (Ortega & Braun, 2013).

Going one step further, one can also optimize for $\rho$ in addition to $\pi$, which essentially means that policies are discouraged to deviate from an optimal prior distribution, leading to the following optimization problem

$$\max_\pi \sum_{s,a} p(s)\pi(a|s)r(s,a) - \frac{1}{\beta}\min_\rho \sum_s p(s)\mathrm{KL}(\pi(\cdot|s)||\rho(\cdot)), \tag{3}$$

where we utilize the fact that only the expected KL penalty depends on $\rho$.

The minimum expected KL relates to the mutual information as follows:

**Proposition 1** (Mutual Information). *Let $I_f$ be a functional, in particular: $I_f(p_X, p_{Y|X}, q_Y) := \sum_x p_X(x)\mathrm{KL}(p_{Y|X}(\cdot|x)||q_Y(\cdot))$, where $p_X(x)$ is the distribution of the input, $p_{Y|X}(y|x)$ is the conditional distribution of the output conditioned on the input, and $q_Y(y)$ a variational distribution of the output. Then, the mutual information [1] is recovered with*

$$I[X,Y] = \min_{q_Y} I_f(p_X, p_{Y|X}, q_Y),$$

*where the optimal variational distribution is $q_Y^\star(y) = \sum_x p_X(x)p_{Y|X}(y|x)$, i.e. the true marginal distribution. See e.g. (Cover & Thomas, 2006, Lemma 10.8.1) for details.*

This allows us to rewrite the problem from Equation (3) as

$$\max_\pi \sum_{s,a} p(s)\pi(a|s)r(s,a) - \frac{1}{\beta}I[S;A], \tag{4}$$

yielding a penalty on the mutual information between states and actions.

Notice that this problem is mathematically equivalent to rate-distortion theory from information-theory (Shannon, 1959) which formulates how to efficiently send information over an information-theoretic channel with limited transmission rate. This framework has also been used to describe decision-making problems with limited information budgets (Sims, 2011; Genewein et al., 2015; Leibfried & Braun, 2015; 2016; Peng et al., 2017; Hihn et al., 2018). In a decision-making context, the agent is considered as information-theoretic channel $\pi(a|s)$ where $s$ is the channel input and $a$ the channel output. The agent aims at maximizing expected reward under the constraint that the information transmission rate is limited, where the transmission rate is given by the mutual-information between states and actions (Cover & Thomas, 2006). Intuitively, this means that the agent has to discard reward-irrelevant information in $s$ to not exceed the limits in information transmission.

In the following section, we generalize the rate-distortion formulation for decision-making to be applicable to a sequential decision-making scenario, i.e. the RL setting. We propose an inference-based formulation where the mutual information arises as a consequence of allowing for optimizing the action-prior distribution.

---

[1] $I(X;Y) := \sum_{x,y} p(x,y)\log\frac{p(x,y)}{p_Y(y)p_X(x)} = \sum_x p_X(x)\mathrm{KL}(p_{Y|X}(\cdot|x)||p_Y(\cdot))$

## 3 Mutual-Information Regularization in RL

In this section, we first derive mutual-information regularization for the RL setting from a variational inference perspective. Subsequently, we derive expressions for the optimal policy and the optimal prior that are useful for constructing a practical algorithm.

### 3.1 Variational Inference RL Formulation with Optimal Action-Priors

The RL problem can be expressed as an inference problem by introducing a binary random variable $R$ that denotes whether the trajectory $\tau := (s_0, a_0, \ldots s_T, a_T)$ is optimal ($R = 1$) or not ($R = 0$). The likelihood of an optimal trajectory can then be expressed as $p(R = 1|\tau) \propto \exp(\sum_{t=0}^{T} r(s_t, a_t))$ (Levine, 2018). We additionally introduce a scaling factor $\beta > 0$ into the exponential, i.e. $p(R = 1|\tau) \propto \exp(\beta \sum_{t=0}^{T} r(s_t, a_t))$. This will allow us to trade off reward and entropy maximization [2]. Next, we can define the posterior trajectory probability assuming optimality, i.e. $p(\tau|R = 1)$. Here we treat $\tau$ as a latent variable with prior probability $p(\tau)$, and we specify the log-evidence as $\log p(R = 1) = \log \int p(R = 1|\tau)p(\tau)d\tau$. We now introduce a variational distribution $q(\tau)$ to approximate the posterior $p(\tau|R = 1)$. This leads to an Evidence Lower BOund (ELBO) of the previous expression (scaled by $\frac{1}{\beta}$) [3]:

$$\frac{1}{\beta} \log p(R = 1) = \frac{1}{\beta} \log \int p(R = 1|\tau)p(\tau)d\tau$$

$$\geq \frac{1}{\beta} \mathbb{E}_{\tau \sim q(\tau)} \left[ \log \frac{p(R = 1|\tau)p(\tau)}{q(\tau)} \right] \tag{5}$$

The generative model is written as $p(\tau) = p(s_0) \prod_{t=0}^{T-1} \rho(a_t)P(s_{t+1}|s_t, a_t)$ and the variational distribution as $q(\tau) = p(s_0) \prod_{t=0}^{T-1} \pi(a_t|s_t)P(s_{t+1}|s_t, a_t)$. The RL problem can now be stated as a maximization of the ELBO w.r.t $\pi$. The maximum entropy RL objective is recovered when assuming a fixed uniform prior distribution over actions, i.e. $\rho(a_t) = \frac{1}{|\mathcal{A}|}$ for all $t$.

We obtain a novel variational RL formulation by introducing an adaptive prior over actions $\rho$. Contrary to maximum entropy RL, where the prior of the generative model is fixed and uniform, here the prior over actions is subject to optimization. Starting from Equation (5) and substituting $p(\tau)$ and $q(\tau)$ we obtain the following ELBO: $\max_{\pi,\rho} \mathbb{E}_q \left[ \sum_{t=0}^{T} \left( r(s_t, a_t) - \frac{1}{\beta} \log \frac{\pi(a_t|s_t)}{\rho(a_t)} \right) \right]$. Since we are interested in infinite horizon problems, we introduce a discount factor and take the limit $\lim_{T \to \infty}$ (Haarnoja et al., 2017). This leads to the optimization objective that we use in our experiments:

$$\max_{\pi,\rho} \mathbb{E}_q \left[ \sum_{t=0}^{\infty} \gamma^t \left( r(s_t, a_t) - \frac{1}{\beta} \log \frac{\pi(a_t|s_t)}{\rho(a_t)} \right) \right], \tag{6}$$

where $0 < \gamma < 1$ is the discount factor. In the following, we show that the the solution for the prior and the policy can be expressed in a concise form giving rise to a novel RL regularization scheme.

### 3.2 Recursion, Optimal Policies and Optimal Priors

Crucial for the construction of a practical algorithm are concise expressions for the optimal policy and the prior. More concretely, the optimal policy takes the form of a Boltzmann distribution weighted by the prior $\rho$. When fixing the policy, the optimal prior is the marginal distribution over actions under the discounted stationary distribution over states. This finding is important to devise a method for efficiently learning an optimal prior in practice.

**Optimal policy for a fixed prior $\rho$:** We start by defining the value function with the information cost as $\mathcal{V}_{\pi,\rho}(s) := \mathbb{E} \left[ \sum_{t=0}^{\infty} \gamma^t \left( r(s_t, a_t) - \frac{1}{\beta} \log \frac{\pi(a_t|s_t)}{\rho(a_t)} \right) |s_0 = s \right]$ where one can show that $\mathcal{V}_{\pi,\rho}$

---

[2] Other authors absorb this scaling factor into the reward, but we keep it as an explicit hyperparameter.

[3] This is obtained by multiplying and dividing by $q(\tau)$ inside the integral and applying Jensen's inequality $f(\mathbb{E}_{x \sim q}[x]) \geq \mathbb{E}_{x \sim q}[f(x)]$ (for any concave function $f$).

satisfies a recursion similar to the Bellman equation:

$$\mathcal{V}_{\pi,\rho}(s) = \mathbb{E}_\pi \left[ r(s,a) - \frac{1}{\beta} \log \frac{\pi(a|s)}{\rho(a)} + \gamma \mathbb{E}_{s'}[\mathcal{V}_{\pi,\rho}(s')] \right]. \tag{7}$$

When considering a fixed $\rho$, the problem of maximizing Equation (7) over the policy can be solved analytically by standard variational calculus (Rubin et al., 2012; Genewein et al., 2015). The optimal policy is then given by

$$\pi^*(a|s) := \frac{1}{Z} \rho(a) \exp(\beta Q_{\pi^*,\rho}(s,a)) \tag{8}$$

with $Z = \sum_a \rho(a) \exp(\beta Q_{\pi^*,\rho}(s,a))$, and the the soft Q-function is defined as

$$Q_{\pi,\rho}(s,a) := r(s,a) + \gamma \mathbb{E}_{s'}[\mathcal{V}_{\pi,\rho}(s')]. \tag{9}$$

Being able to write the optimal policy in this way as a function of Q-values is needed in order to estimate the optimal prior as we show next.

**Optimal prior for a fixed policy:** In order to solve for the optimal prior, we rewrite the problem in Equation (6) as

$$\max_{\pi,\rho} \sum_{t=0}^\infty \sum_s \gamma^t \nu_t(s) \sum_a \pi(a|s) \left( r(s,a) - \frac{1}{\beta} \log \frac{\pi(a|s)}{\rho(a)} \right),$$

where we have defined the marginal distribution over states at time $t$ as

$$\nu_t(s) := \sum_{s_0, a_0, \ldots, s_{t-1}, a_{t-1}} p(s_0) \left( \prod_{t'=0}^{t-2} \pi(a_{t'}|s_{t'}) P(s_{t'+1}|s_{t'}, a_{t'}) \right) \pi(a_{t-1}|s_{t-1}) P(s|s_{t-1}, a_{t-1}). \tag{10}$$

For fixed $\pi$, we eliminate the max operator for $\pi$ and all components that do not depend on $\rho$:

$$\max_\rho -\frac{1}{\beta} \sum_{t=0}^\infty \sum_s \gamma^t \nu_t(s) \mathrm{KL}(\pi(\cdot|s)||\rho(\cdot)).$$

Swapping the sums and letting $p(s) := \sum_{t=0}^\infty \gamma^t \nu_t(s)$ be the unnormalized discounted marginal distribution over states, we obtain $\max_\rho -\frac{1}{\beta} \sum_s p(s) \mathrm{KL}(\pi(\cdot|s)||\rho(\cdot))$. The solution to the latter, $\rho^\star(a) = \frac{\sum_s p(s)\pi(a|s)}{\sum_{s,a} p(s)\pi(a|s)}$, can easily be obtained by adding the constraint that the action-prior is a valid distribution (i.e., $\sum_a \rho(a) = 1$ and $\rho(a) > 0 \; \forall a$), and using the method of Lagrange multipliers and standard variational calculus. The connection to the mutual information becomes clear when plugging $\rho^\star$ back into the objective yielding $-k \cdot \frac{1}{\beta} I(S, A)$ scaled by a positive constant $k$ (because $p(s)$ is not normalized) that can be absorbed into $\beta$. Additionally, we also formalize the connection to the stationary mutual-information for the limit case of $\gamma \to 1$ in the Appendix.

With the form of the optimal prior for a fixed policy at hand, one can easily devise a stochastic approximation method (e.g. $\rho_{i+1}(a) = (1 - \alpha_\rho)\rho_i(a) + \alpha_\rho \pi(a|s)$ with $\alpha_\rho \in [0, 1]$ and $s \sim p(\cdot)$ to estimate the optimal $\rho$ using the current estimate of the optimal policy from Equation (8). We note that here we sample from the undiscounted distribution over states instead, rather than the true discounted state-distribution in the equation above. This is a common practice used in actor-critic RL that results in a biased estimator of the objective (Thomas, 2014).

## 4 MIRL: A PRACTICAL MUTUAL INFORMATION RL ALGORITHM

In this section, we present the MIRL agent. We focus on the tabular setting first and then port our algorithm to high-dimensional state spaces that require parametric function approximators.

### 4.1 MIRL FOR TABULAR Q-LEARNING

Our tabular MIRL agent is a modification of an ordinary Q-learning agent with a different update scheme for Q-values. In parallel to updating Q-values, the MIRL agent needs to update the prior $\rho$ as well as the parameter $\beta$. The behavioral policy $\pi$ is also different and utilizes soft Q-values. This is outline in more detail below.

**Prior Updates:** We approximate the optimal prior by employing the following update equation,

$$\rho_{i+1}(a) = (1 - \alpha_\rho)\rho_i(a) + \alpha_\rho \pi_i(a|s_i) \tag{11}$$

where $s_i \sim \nu_i(\cdot)$ and $\alpha_\rho$ is a learning rate. Assuming a fixed policy, it is easy to show that this iterative update converges to $\rho_\pi(a) = \sum_s \mu_\pi(s)\pi(a|s)$, thus estimating correctly the optimal prior.

**Q-function Updates:** Concurrently to learning the prior, MIRL updates the tabular Q-function as

$$Q(s,a) \leftarrow Q(s,a) + \alpha_Q \left( (T^\rho_{\text{soft}} Q_{\bar\theta})(s,a,s') - Q(s,a) \right) \tag{12}$$

where $\alpha_Q$ is a learning rate and $T^\rho_{\text{soft}} Q$ is the empirical soft-operator defined as $(T^\rho_{\text{soft}} Q)(s,a,s') :=$ $r(s,a) + \gamma \frac{1}{\beta} \log \sum_{a'} \rho(a') \exp(\beta Q(s',a'))$. Importantly, this operator differs from other soft operators arising when employing entropy regularization. Entropy regularization assumes a fixed uniform prior, whereas in our case, the optimal prior is estimated in the course of learning.

**Behavioural policy:** Since the Q-function can be learned off-policy, the experience samples can conveniently be drawn from a behavioural policy $\pi_b$ different from the current estimate of the optimal policy. As such, the behavioural policy used in our experiments is similar in spirit to an $\epsilon$-greedy policy but it better exploits the existence of the estimated optimal prior when both exploring and exploiting. When exploring, MIRL's behavioural policy samples from the current estimate of the optimal prior $\rho_i$ which has adjusted probabilities, in contrast to vanilla $\epsilon$-greedy that samples all actions with equal frequency. Additionally, when exploiting, MIRL selects the maximum probability action that depends not only on the Q-values but also on the current estimate of the optimal action-prior, instead of selecting the action with highest Q-value as in traditional $\epsilon$-greedy. More formally, given a random sample $u \sim \text{Uniform}[0,1]$ and epsilon $\epsilon$, the action $a_i$ is obtained by

$$a_i = \begin{cases} \arg\max_a \pi_i(a|s_i), & \text{if } u > \epsilon \\ a \sim \rho_i(\cdot) & \text{if } u \leq \epsilon, \end{cases}$$

where $\pi_i(a|s) = \frac{1}{Z}\rho_i(a)\exp(\beta_i Q_i(s,a))$, see Equation (8).

**Parameter $\beta$ Updates:** The parameter $\beta$ can be seen as a Lagrange multiplier that quantifies the magnitude of penalization for deviating from the prior. As such, a small fixed value of $\beta$ would restrict the class of available policies and evidently constrain the asymptotic performance of MIRL. In order to remedy this problem and obtain better asymptotic performance, we use the same adaptive $\beta$-scheduling over rounds $i$ from (Fox et al., 2016) in which $\beta_i$ is updated linearly according to $\beta_{i+1} = c \cdot i$ with some positive constant $c$. This update favours small values of $\beta$ at the beginning of training and large values towards the end of training when the error over Q-values is small. Therefore, towards the end of training when $\beta$ is large, MIRL recovers ordinary Q-learning without a constraint. This ensures that the asymptotic performance of MIRL is not hindered.

## 4.2 MIRL WITH PARAMETRIC FUNCTION APPROXIMATORS

For parametric function approximators, the scheme for updating the prior and the behavioural policy is the same as in the tabular setting but Q-function updates and $\beta$-scheduling need to be adjusted for high-dimensional state spaces. The pseudocode of our proposed algorithm is outlined in Algorithm 1 and follows standard literature for parametric value learning, see e.g. Mnih et al. (2015).

**Q-function Updates:** Q-function parameters are obtained by minimizing the following loss

$$L(\theta, \rho) := \mathbb{E}_{s,a,r,s' \sim \mathcal{M}} \left[ \left( (T^\rho_{\text{soft}} Q_{\bar\theta})(s,a,s') - Q_\theta(s,a) \right)^2 \right] \tag{13}$$

where $\mathcal{M}$ is a replay memory (Mnih et al., 2015), $Q_{\bar\theta}$ is a target network that is updated after a certain number of training iterations and $T^\rho_{\text{soft}} Q$ is the empirical soft-operator from the tabular setting, here repeated for convenience $(T^\rho_{\text{soft}} Q)(s,a,s') := r(s,a) + \gamma \frac{1}{\beta} \log \sum_{a'} \rho(a') \exp(\beta Q(s',a'))$.

**Parameter $\beta$ Updates:** We use the same adaptive $\beta$-scheduling from Leibfried et al. (2018) in which $\beta_i$ is updated according to the inverse of the empirical loss of the Q-function, i.e. $\beta_{i+1} = (1 - \alpha_\beta)\beta_i + \alpha_\beta \left( \frac{1}{L(\theta_i, \rho_{i+1})} \right)$. This provides more flexibility than the linear scheduling scheme from the tabular setting, more suitable for high-dimensional state spaces where it is impossible to visit all state-action pairs.

---

**Algorithm 1** MIRL

---

1: **Input:** the learning rates $\alpha_\rho$, $\alpha_Q$ and $\alpha_\beta$, a Q-network $Q_\theta(s, a)$, a target network $Q_{\bar{\theta}}(s, a)$, a behavioural policy $\pi_b$, an initial prior $\rho_0$ and parameters $\theta_0$ at $t = 0$.
2: **for** $i = 1$ to $N$ iterations **do**
3:     Get environment state $s_i$ and apply action $a_i \sim \pi_b(\cdot|s_i)$
4:     Get $r_i, s_{i+1}$ and store $(s_i, a_i, r_i, s_{i+1})$ in replay memory $\mathcal{M}$
5:     Update prior $\rho_{i+1}(\cdot) = \rho_i(\cdot)(1 - \alpha_\rho) + \alpha_\rho \pi_i(\cdot|s_i)$
6:     **if** $i$ mod update frequency $== 0$ **then**
7:         Update Q-function $\theta_{i+1} = \theta_i - \alpha_Q \nabla_\theta L(\theta, \rho_{i+1})|_{\theta_i}$ according to Equation (13)
8:         Update parameter $\beta_{i+1} = (1 - \alpha_\beta)\beta_i + \alpha_\beta \left( \frac{1}{L(\theta_i, \rho_{i+1})} \right)$
9:     **end if**
10: **end for**

---

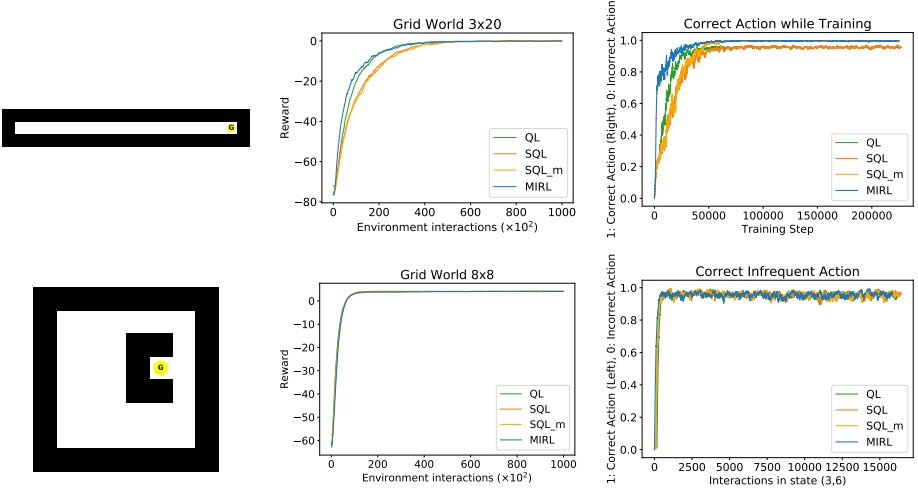

Figure 1: Grid world experiments. **Left column:** The top shows a corridor grid world with $3 \times 20$ cells where the goal is on the right. The bottom shows an $8 \times 8$ grid world where an important action (left) has to be made exactly once to arrive at the goal. **Middle column:** Evaluation of Q-learning (QL), SQL with standard uniform exploration and with marginal exploration (SQL_m), and MIRL. We clearly see that MIRL outperforms the baselines on the corridor, and is comparable to the baselines on the $8 \times 8$ world. **Right column:** We see that MIRL is able to identify the correct action (go right) faster than the baselines in the corridor (top). The bottom reports how having infrequent but important actions does not affect the performance of MIRL.

## 5 EXPERIMENTS

We evaluate our MIRL agent both in the tabular setting using a grid world domain, and in the parametric function approximator setting using the Atari domain.

### 5.1 GRID WORLD

As an intuitive example, we evaluate our method in a grid world domain where the agent has to reach a goal. Reaching the goal gives a reward of 9 but each step yields a reward of $-1$. After the end of an episode (when reaching the goal), the agent's location is randomly re-sampled uniformly over the state space. We compare against two baselines, Q-learning without any regularization, and SQL (Fox et al., 2016) which employs entropy regularization with the dynamic $\beta$-scheduling scheme outlined earlier. We train the agents for $2.5 \cdot 10^5$ environment steps following the procedure outlined in Fox et al. (2016). Both SQL and MIRL update the Lagrange multiplier $\beta$ over time by using a linear scheduling scheme with a constant $c = 10^{-3}$. MIRL additionally updates the estimate of the optimal prior by using a learning rate $\alpha_\rho = 2 \cdot 10^{-3}$. In all experiments, we use an adaptive

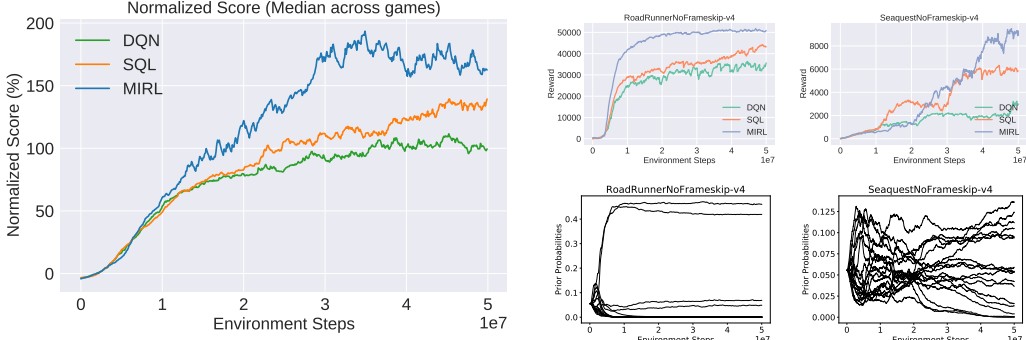

Figure 2: **Left panel:** Median normalized score across 19 Atari games. Comparison between our method mutual information RL (MIRL), SQL and DQN, demonstrating MIRL's superior performance. **Right panels:** Top figures show the raw score for 2 example games reporting MIRL's superior performance on RoadRunner and Seaquest. The bottom plots show the evolution of the estimated prior over actions. For RoadRunner the prior converges to stable values during training. In Seaquest, the algorithm seems not to have converged yet after 50 million environment steps which is why the prior probabilities have not converged yet either (however, the formation of separate trajectory clusters towards the end of training indicates the ongoing process of convergence). See Appendix for details and plots for all environments. The curves are smoothed with an exponential moving average with effective window size of $10^6$ environment steps.

learning rate for Q-values $\alpha_Q = n(s,a)^{-\omega}$ that depends on the state-action-visitation frequencies $n(s,a)$ (Fox et al., 2016). See Appendix for further details.

During the training of each algorithm, snapshots of the Q-tables (and the estimate of the prior in the case of MIRL) are stored every 100 environment steps for evaluation. The evaluation for a single snapshot is conducted by running the policy for 30 episodes lasting at most 100 environment steps. The epsilon value when in evaluation mode is set to $\epsilon = 0.05$ (same as in training). Every individual experiment is repeated with 10 different initial random seeds and results are averaged across seeds.

Figure 1 summarizes our grid world experiments on two instances: a corridor where the goal is to the right, and a square world where one important action has to be executed exactly once. In the corridor, MIRL clearly outperforms competing approaches (Q-learning and SQL), whereas in the square world, MIRL attains comparable performance as the baselines. Note that these results remain valid when equipping SQL with the same marginal exploration scheme as MIRL.

## 5.2 ATARI

We conduct experiments on 19 Atari games (Brockman et al., 2016) with Algorithm 1 (MIRL), and compare against DQN (Mnih et al., 2015) and SQL (Haarnoja et al., 2018c) with a dynamic $\beta$-scheduling scheme based on loss evolution that leads to improved performance over vanilla SQL with fixed $\beta$ (Leibfried et al., 2018). The three algorithms use a neural network for the estimation of Q-values as in (Mnih et al., 2015). The network receives as an input the state $s$ which is composed of the last four frames of the game with some extra pre-processing (see Appendix), and it outputs a vector of Q-values, i.e. one value for each valid action. We train the network for $5 \cdot 10^7$ environment steps, where a training iteration is performed every four steps. The target network $Q_{\bar{\theta}}$ is updated every $10^4$ training iterations. Both SQL and MIRL update the Lagrange multiplier $\beta$ over time by using an exponential moving average of the inverse loss [4] (Leibfried et al., 2018) with $\alpha_\beta = 3 \cdot 10^{-5}$. In addition, MIRL updates the estimate of the optimal prior by using a learning rate $\alpha_\rho = 5 \cdot 10^{-5}$. Additional details can be found in the Appendix.

For evaluation, we create snapshots of the network agents every $10^5$ environment steps. Evaluating a single snapshot offline is done by running the policy for 30 episodes that last at most $4.5 \cdot 10^3$ environment steps but terminate earlier in case of a terminal event. When evaluating the agents, the

---

[4]Pracitcally, we replace the squared loss with the Huber loss Mnih et al. (2015).

epsilon value is $\epsilon = 0.05$, whereas in training $\epsilon$ is linearly annealed over the first $10^6$ steps (Mnih et al., 2015) from $1.0$ to $0.1$.

To summarize the results across all games, we normalize the episodic rewards obtained in the evaluation. The normalized episodic rewards are computed as follows $z_{\text{normalized}} = \frac{z - z_{\text{random}}}{z_{\text{human}} - z_{\text{random}}} \cdot 100\%$, where $z$ stands for the score obtained from our agent at test time, $z_{\text{random}}$ stands for the score that a random agent obtains and $z_{\text{human}}$ for the score a human obtains. Random and human scores are taken from Mnih et al. (2015) and Van Hasselt et al. (2016). As seen in Figure 2, our algorithm significantly outperforms the baselines in terms of the median normalized score. In particular, after 50 million interactions we obtain about $30\%$ higher median normalized score compared to SQL and $50\%$ higher score compared to DQN. MIRL attains the final performance of SQL in about half the amount of interactions with the environment and, similarly, it attains DQN's final performance in about five times less interactions.

In Table 1 we show the comparison between best-performing agents for all the

| Game | DQN (%) | SQL (%) | MIRL (%) |
|------|---------|---------|----------|
| Alien | **101.58** | 51.02 | 40.23 |
| Assault | 250.61 | 283.62 | **357.40** |
| Asterix | 166.32 | 242.73 | **330.19** |
| Asteroids | **9.74** | 8.57 | 7.80 |
| BankHeist | 97.12 | 94.62 | **166.26** |
| BeamRider | 99.16 | 113.64 | **117.21** |
| Boxing | 2178.57 | 2283.33 | **2338.89** |
| ChopperCommand | **72.71** | 26.37 | 65.03 |
| DemonAttack | 350.95 | 451.78 | **469.30** |
| Gopher | 474.18 | **538.87** | 429.44 |
| Kangaroo | 351.48 | 393.16 | **405.9** |
| Krull | 843.16 | 886.68 | **1036.04** |
| KungFuMaster | 122.14 | **142.04** | 121.41 |
| Riverraid | 77.21 | **109.37** | 76.02 |
| RoadRunner | 548.90 | 613.62 | **695.88** |
| Seaquest | 21.95 | 36.00 | **64.86** |
| SpaceInvaders | 166.62 | **200.38** | 164.79 |
| StarGunner | 653.44 | **681.12** | 574.89 |
| UpNDown | 183.19 | 230.82 | **394.21** |
| Mean | 356.26 | 388.83 | **413.46** |

Table 1: Mean Normalized score in 19 Atari games for DQN, SQL and our approach MIRL.

environments, where the best-performing agent is the agent that achieves the best score in evaluation mode considering all snapshots. Although this measure is not very robust, we include it since it is a commonly reported measure of performance in the field. MIRL outperforms the other baselines in 11 out of 19 games compared to SQL and DQN that are best on 5 and 3 games respectively.

In Figure 3, we conduct additional experiments on a subset of eight Atari games comparing MIRL and DQN with two different ablations of SQL: one ablation using uniform exploration and another ablation using the same marginal exploration scheme as MIRL (denoted SQL_m). These experiments confirm the importance of the difference in values between MIRL and SQL rather than the difference in the exploration protocol.

## 6 RELATED WORK

The connection between reinforcement learning and inference is well established (Dayan & Hinton, 1997; Levine, 2018). Several authors have proposed RL algorithms based on optimizing the ELBO in Equation (5). In policy search, a popular approach is to optimize the lower bound using an iterative procedure similar to expectation maximization (Deisenroth et al., 2013). Different approximations can then be used for the trajectory distributions, resulting in different algorithms (Kober & Peters, 2009; Peters et al., 2010; Hachiya et al., 2011). The recent maximum a posteriori policy optimisation (MPO) (Abdolmaleki et al., 2018) framework uses a similar approach and combines an expectation maximization style update with off-policy estimation of a regularized Q-function. A key difference between MPO and our method is that MPO treats the approximate distribution $q(\tau)$ as an auxiliary distribution used to optimize the policy that generates $p(\tau)$. In our method, as well as in the maximum entropy based methods discussed below, we optimize the policy used to generate $q(\tau)$, while using the distribution $p(\tau)$ generated by the prior as an auxiliary distribution. While both approaches can be related to optimizing the ELBO, they can be shown to optimize different versions of the KL constraint on the target policy (Levine, 2018).

Maximum entropy reinforcement learning represents another family of inference-based RL methods. This formulation can be derived from the same evidence lower bound in Equation (5), by fixing the generative policy for $p(\tau)$ to be a uniform prior. Maximum entropy RL has been derived under

different conditions in many different settings. Ziebart et al. (2008) proposed maximum entropy learning for inverse reinforcement learning problems. Several authors have applied the same principles for trajectory optimization (Kappen, 2005; Todorov, 2008; Levine & Koltun, 2013). These maximum entropy methods typically assume the availability of some sort of transition model. More recently, maximum entropy learning has also been studied in model-free settings by introducing alternative soft-operators (Asadi & Littman, 2017) or soft Q-learning approaches (Rawlik et al., 2012; Fox et al., 2016; Haarnoja et al., 2017). Soft Q-learning learns a softened value function by replacing the hard maximum operator in the Q-learning update with a softmax operator. Several authors have discussed the benefits of this approach and provided generalizations under linear programming formulations (Neu et al., 2017). In particular, Fox et al. (2016) and Haarnoja et al. (2017) show that maximum entropy learning improves exploration and robustness. Furthermore, Haarnoja et al. (2018b) show that the resulting policies are composable and can be used to directly build solutions to unseen problems. Additionally, entropy-regularization has shown to be crucial to prove convergence guarantees on value learning with non-linear function approximators (Dai et al., 2018). The soft Q-learning framework has also been used in actor-critic settings (Haarnoja et al., 2018c) and to show a connection between value-based and policy gradient methods (Schulman et al., 2017; Nachum et al., 2017). The method has also been extended to hierarchical settings (Florensa et al., 2017; Haarnoja et al., 2018a). In the multi-task setting, the Distral framework (Teh et al., 2017) combines entropy regularization with an additional KL regularization used to transfer knowledge between tasks.

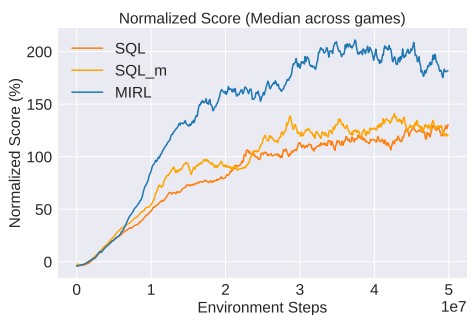

The mutual information, central to our approach, is a basic quantity in information theory to measure the statistical dependence between two random variables. Machine learning applications that use the mutual information are numerous including the information-bottleneck method (Tishby et al., 1999), rate-distortion theory (Cover & Thomas, 2006; Tishby & Polani, 2011), clustering (Still & Bialek, 2004) and curiosity driven exploration (Still & Precup, 2012).

Figure 3: Normalized score for eight games comparing MIRL against standard SQL and a modified version of SQL that explores with the marginal distribution over actions (SQL_m). The exploration method slightly improves SQL but not sufficiently enough to achieve MIRL's performance. See individual plots and games in the Appendix.

## 7 DISCUSSION AND CONCLUSION

Using a variational inference perspective, we derived a novel RL objective that allows optimization of the prior over actions. This generalizes previous methods in the literature that assume fixed uniform priors. We show that our formulation is equivalent to applying a mutual-information regularization and derive a novel algorithm (MIRL) that learns the prior over actions. We demonstrate that MIRL significantly improves performance over SQL and DQN.

We recognize that our approach might fail under certain conditions. For example, in the case when there is an action that is useful only once (similar to our $8 \times 8$ grid world example) but is most of the times penalized with a negative reward. When the negative reward is too strong, MIRL might assign very low probability to that action and never explore it. However, this problem might be alleviated by a weighted mixing of our exploration policy with a uniform distribution.

An interesting direction for future work is to investigate the convergence properties of the alternating optimization problem presented here. We believe that, at least in the tabular case, the framework of stochastic approximation for two timescales (Borkar, 2009) is sufficient to prove convergence. On the experimental side, one could also investigate how our approach can be combined with the Rainbow framework (Hessel et al., 2017) which is the current state of the art in performance.

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

## A    APPENDIX

### A.1    CONNECTION TO MUTUAL INFORMATION FOR $\gamma \to 1$.

The goal of this section is to show that when $\gamma \to 1$, Equation (6) can be expressed as the following average-reward formulation (Puterman, 1994) with a constraint on the stationary mutual information

$$\max_{\pi} \mathbb{E}_{s \sim \mu_{\pi}} \left[ \sum_a \pi(a|s) r(s,a) \right] \quad \text{s.t. } I_f(\mu_{\pi}, \pi, \rho_{\pi}) \leq C, \tag{14}$$

where $\mu_{\pi}$ and $\rho_{\pi}$ are the stationary distributions induced by the policy $\pi$ over states and actions, respectively, and thus $I_f(\mu_{\pi}, \pi, \rho_{\pi})$ is defined as the stationary mutual-information. Note that for a fixed stationary distribution over states, this problem coincides exactly with the well-known rate-distortion problem (Cover & Thomas, 2006).

We start by expressing (6) as a constrained problem

$$\max_{\pi} \mathbb{E}_q \left[ \sum_{t=0}^{\infty} \gamma^t r(s_t, a_t) \right] \quad \text{s.t. } \min_{\rho} \sum_{t=0}^{\infty} \gamma^t I_f(\nu_t, \pi, \rho) \leq K(\gamma), \tag{15}$$

where $K(\gamma) := \frac{C}{1-\gamma}$ (although we have set $K(\cdot)$ as a function of $\gamma$, it is without loss of generality since we can always obtain a desired $K(\cdot)$ by choosing an appropriate $C$ for a given $\gamma$) and the marginal probability of state $s_t$ at time $t$ following the Markovian dynamics is written as in Equation (10).

A standard result in the MDP literature (Bertsekas, 1995) is that

$$\arg\max_\pi \lim_{\gamma\to 1}(1-\gamma)\mathbb{E}_q\left[\sum_{t=0}^\infty \gamma^t r(s_t,a_t)\right] \iff \arg\max_\pi \mathbb{E}_{s\sim\mu_\pi}\left[\sum_a \pi(a|s)r(s,a)\right]$$

which basically says that the optimal policy for the limit $\gamma \to 1$ of an infinite horizon problem is equivalent to the average reward formulation. Now it only remains to make explicit a similar equivalence on the constraint.

We rewrite the constraint by multiplying on both sides by $(1-\gamma)$ assuming $\gamma \in (0,1)$

$$\min_\rho \sum_{t=0}^\infty \gamma^t I_f(\nu_t,\pi,\rho) \le K(\gamma) \iff \min_\rho (1-\gamma)\sum_{t=0}^\infty \gamma^t I_f(\nu_t,\pi,\rho) \le C.$$

Taking the limit $\gamma \to 1$ in the last inequality and interchanging the limit and the min operators [5], we obtain the constraint $\min_\rho \lim_{\gamma\to 1}(1-\gamma)\sum_{t=0}^\infty \gamma^t I_f(\nu_t,\pi,\rho) \le C$. Then, we see the connection between the last inequality and the constraint on Equation (14) using the following Proposition 2.

**Proposition 2.** *Let $\mu_\pi(s)$ and $\rho_\pi(a)$ be the stationary distribution over states and actions under policy $\pi$ according to Definitions (1) and (2). Then the stationary mutual information defined as $I_f(\mu_\pi,\pi,\rho_\pi) := \min_\rho I_f(\mu_\pi\pi,\rho)$ can also be written as*

$$I_f(\mu_\pi,\pi,\rho_\pi) = \min_\rho \lim_{\gamma\to 1}(1-\gamma)\sum_{t=0}^\infty \gamma^t I_f(\nu_t,\pi,\rho) \qquad (16)$$

*where $\nu_t(s)$ is defined as in (10).*

*Proof.* Following similar steps as in (Bertsekas, 1995, p.186) we have

$$\begin{aligned}
I_f(\mu_\pi,\pi,\rho_\pi) &= \min_\rho \lim_{N\to\infty}\frac{1}{N}\sum_{t=0}^{N-1} I_f(\nu_t,\pi,\rho) \\
&= \min_\rho \lim_{N\to\infty}\lim_{\gamma\to 1}\frac{\sum_{t=0}^{N-1}\gamma^t I_f(\nu_t,\pi,\rho)}{\sum_{t=0}^N \gamma^t} \\
&= \min_\rho \lim_{\gamma\to 1}\lim_{N\to\infty}\frac{\sum_{t=0}^N \gamma^t I_f(\nu_t,\pi,\rho)}{\sum_{t=0}^N \gamma^t} \\
&= \min_\rho \lim_{\gamma\to 1}(1-\gamma)\sum_{t=0}^\infty \gamma^t I_f(\nu_t,\pi,\rho),
\end{aligned}$$

where we used Proposition 3 (shown next) in the first equality and where the limits in the third equality can be interchanged due to the monotone convergence theorem. $\qquad\square$

**Proposition 3.** *Let $\mu_\pi(s)$ and $\rho_\pi(a)$ be the stationary distribution over states and actions under policy $\pi$ according to Definitions (1) and (2). Then the stationary mutual information defined as $I_f(\mu_\pi,\pi,\rho_\pi)$ can also be written as*

$$I_f(\mu_\pi,\pi,\rho_\pi) = \min_\rho \lim_{N\to\infty}\frac{1}{N}\sum_{t=0}^{N-1}\sum_{s_t} \nu_t(s_t)KL(\pi(\cdot|s_t)||\rho(\cdot)), \qquad (17)$$

*where $\nu_t(s)$ is defined as in (10).*

---

[5]We can interchange the operators because the l.h.s. of the inequality is finite for any $0 < \gamma < 1$.

*Proof.* Let $\mathcal{I}_\rho(s) := \text{KL}(\pi(\cdot|s)||\rho(\cdot))$ and $\mathcal{I}_{\rho_\pi}(s) := \text{KL}(\pi(\cdot|s)||\rho_\pi(\cdot))$. Note that both previous quantities are bounded for all $t$. Then

$$\min_\rho \lim_{N\to\infty} \frac{1}{N} \sum_{t=0}^{N-1} \sum_s \nu_t(s)\mathcal{I}_\rho(s) =$$

$$= \min_\rho \lim_{N\to\infty} \frac{1}{N} \left( \sum_{t=0}^{K} \sum_s \nu_t(s)\mathcal{I}_\rho(s) + \sum_{t=K+1}^{N-1} \sum_s \nu_t(s)\mathcal{I}_\rho(s) \right)$$

$$= \min_\rho \lim_{N\to\infty} \frac{1}{N} \sum_{t=K+1}^{N-1} \sum_s \nu_t(s)\mathcal{I}_\rho(s)$$

$$= \min_\rho \lim_{N\to\infty} \frac{1}{N} \sum_{t=K+1}^{N-1} \sum_s \mu_\pi(s)\mathcal{I}_\rho(s)$$

$$= \min_\rho \sum_s \mu_\pi(s)\mathcal{I}_\rho(s) \lim_{N\to\infty} \frac{1}{N} \sum_{t=K+1}^{N-1} 1$$

$$= \sum_s \mu_\pi(s)\mathcal{I}_\mu(s) \lim_{N\to\infty} \frac{N-K}{N}$$

$$= \sum_s \mu_\pi(s)\mathcal{I}_\mu(s)$$

$$= I_f(\mu_\pi, \pi, \rho_\pi),$$

where we assumed that $\nu_t(s) = \mu_\pi(s)$ for all $t > K$ and finite but large enough $K$. □

Since in practice we use a discount factor $\gamma \lessgtr 1$, our original problem formulation in (6) can be seen as an approximation to the problem with stationary mutual-information constraints in (14).

The conclusion of this section is that we have established a clear link between the ELBO with optimizable priors and the average reward formulation with stationary mutual-information constraints.

## A.2 HYPERPARAMETERS

Here we describe the hyperparameters used for both the Grid World experiments (see Table 2) and the Atari experiments (see Table 3).

| Parameter | Value |
|:---:|:---:|
| $\gamma$ | 0.99 |
| $\alpha_\rho$ | $2 \cdot 10^{-3}$ |
| $\omega$ | 0.8 |

Table 2: Hyperparameters for tabular experiments.

| Parameter | Value |
|---|---|
| Frame size | $[84, 84]$ |
| Frame skip | 4 |
| History frames (in $s$) | 4 |
| Reward clipping | $\{-1, 0, +1\}$ |
| Max environment steps | 27000 |
| Target update frequency (train. steps) | 10000 |
| Training update frequency (env. steps) | 4 |
| Batch size | 32 |
| Memory capacity | $10^6$ |
| $\gamma$ | 0.99 |
| $\alpha_\rho$ | $2 \cdot 10^{-6}$ |
| $\alpha_Q$ | $2 \cdot 10^{-5}$ |
| $\alpha_\beta$ | $3.3 \cdot 10^{-6}$ |
| $\beta_0$ | 0.01 |

Table 3: Hyperparameters for Atari experiments.

## A.3 SPECIFIC PLOTS FOR INDIVIDUAL GAMES

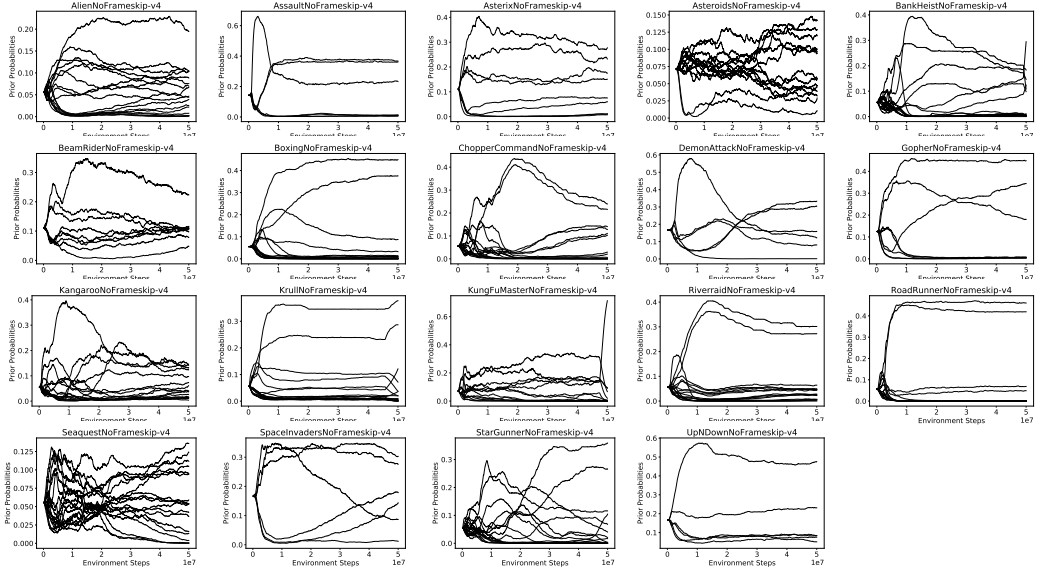

Figure 4: Prior Evolution for all games. We can see that MIRL's prior has fully converged for some games whereas for other games it is still about to converge.

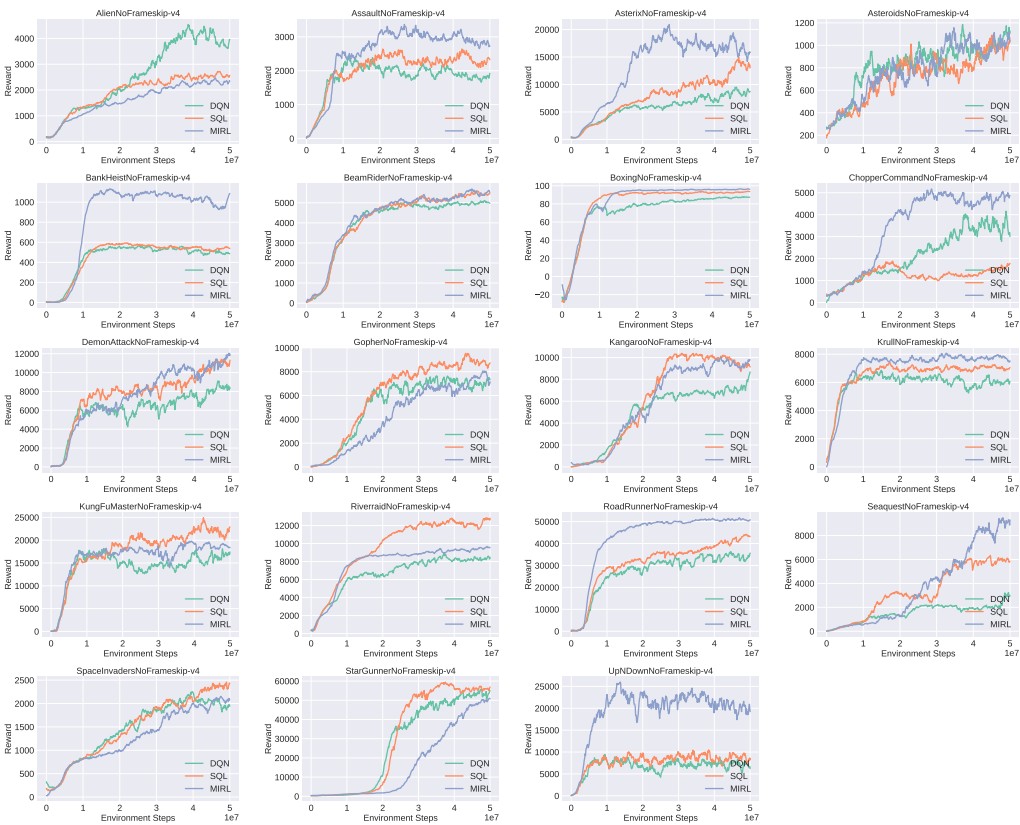

Figure 5: Scores for all games on the evaluation snapshots.

## A.4 ABLATION STUDY

In the ablation study summarized in Figure 3, we show how marginal exploration affects SQL. In Figure 6, we show the same plot for individual games. We clearly see that the marginal exploration improves performance, but is not the defining factor for all the improvements obtained by MIRL.

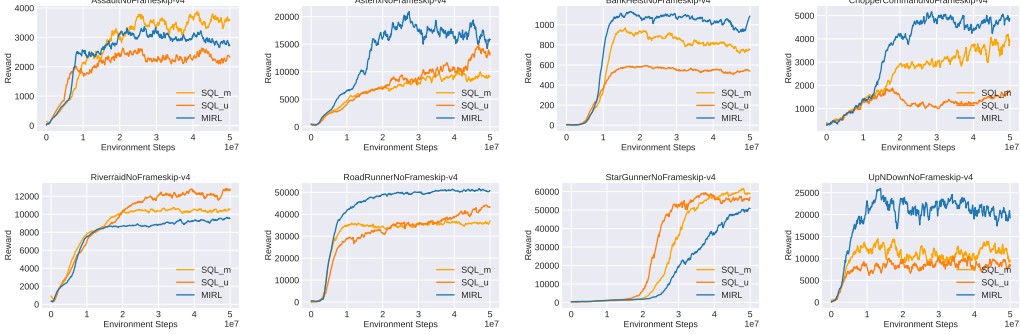

Figure 6: Comparison between standard SQL (SQL_u), SQL with marginal exploration (SQL_m), and MIRL.

## A.5 EVOLUTION OF THE LAGRANGE MULTIPLIER

In Figure 7, we show the evolution of $\beta$ over time (environment steps) for the MIRL agent. As we can see, the $\beta$-values usually start at a high value (not shown for visual reasons) and typically go down and stabilize at some value. At first sight, this might be seen as a negative side effect since

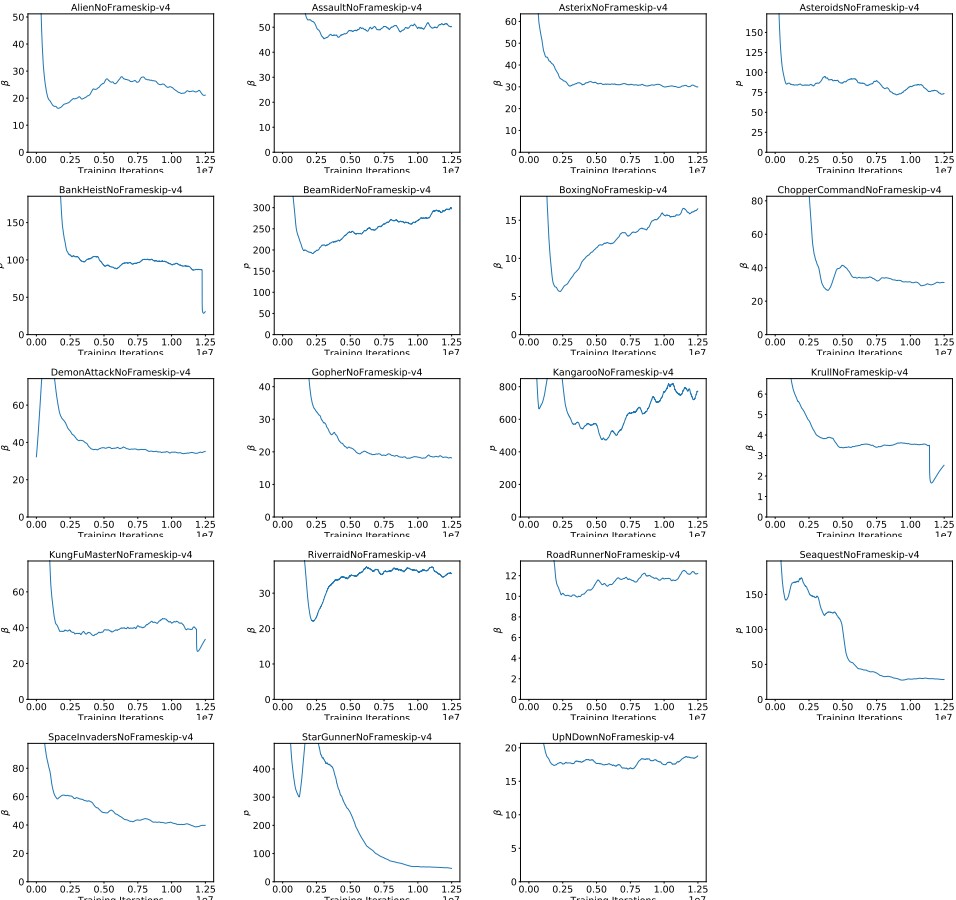

Figure 7: Beta evolution over time.

lower $\beta$ values imply a stronger constraint. However, we note that the constraint is highly dependent on the scale of the reward (or its sparsity), and therefore, the $\beta$ value is not meaningful without a proper specification of this reward scaling.

Consequently, given that $\beta$-values are not meaningful here, we propose to instead show the multiplication of $\beta$ times the current maximum Q-value estimates denoted as $\beta \times \max Q$. Note that $\beta Q(s, a)$ appears on the exponential term of the policy, i.e. $\pi(a|s) = \frac{1}{Z}\rho(a)\exp(\beta Q(s, a))$, and therefore, is the term that shapes the deviation from the action-prior distribution. Additionally, the Q-values serve us as proper scaling for each game and account also for the learning of the agent. In particular, while the agent is learning and increasing its reward acquisition, the Q-values are going to be higher, thus, effectively needing a smaller $\beta$ to shape the policy probabilities.

On Figure 8, we show the evolution of the term $\beta \times \max Q$ for all the games. As we can see for the majority of games, $\beta \times \max Q$ increases over time or has high value. This is important since a high value denotes that the policy is highly affected by this term.

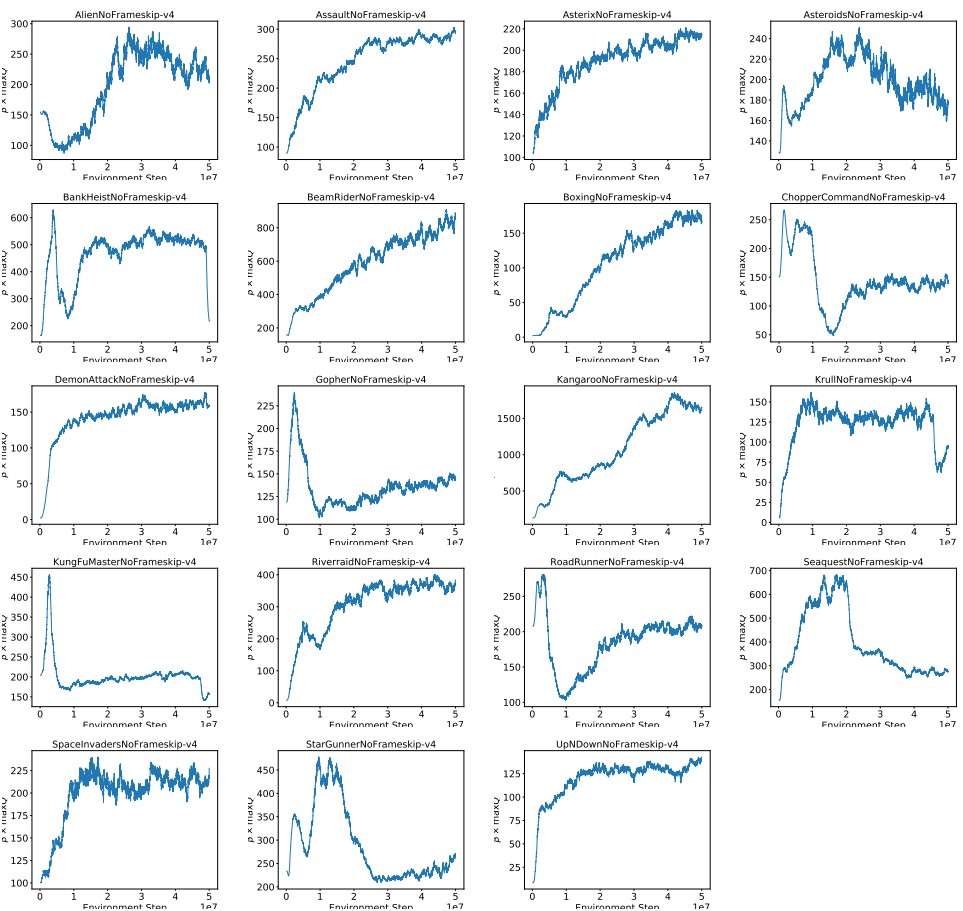

Figure 8: Evolution of $\beta \times \max Q$ over time while training. Specifically, for an environment step $i$, we compute the $\beta_i \max_a Q_{\theta_i}(s_i, a)$, where $\beta_i$ is the current $\beta$-value, $Q_{\theta_i}$ the current approximation of Q and $s_i$ is the state at the step $i$.

