# OpenReview forum: "Soft Q-Learning with Mutual-Information Regularization"
_ICLR.cc/2019/Conference_

### Official Review · AnonReviewer1 · 2018-10-31
**Interesting, but motivation and experiments need improvements**

**Rating:** 6
**Confidence:** 3

**Review:**

The authors take the control-as-inference viewpoint and learn a state-independent prior (which is typically held fixed). They claim that this leads to better exploration when actions have different importance. They relate this objective to a mutual information constrained RL objective in a limiting case. They then propose a practical algorithm, MIRL and compare their algorithm against DQN and Soft Q-learning (SQL) on 19 Atari games and demonstrate improvements over both.

Generally I found the idea interesting and at a high level the deficiency of entropy regularization makes sense. However, I had great trouble understanding the reasoning behind their method and did not find the connection to mutual information helpful. Furthermore, I had a number of questions about the experiments. If the authors can clarify their motivation and reasoning and strengthen the experiments, I'd be happy to raise my score.

In Sec 3.1, why is it sensible to optimize the prior? Can the authors give intuition for maximizing \log p(R = 1) wrt to the prior? This is critical for justifying their approach. Currently, the authors provide a connection to MI, but don't explain why this matters. Does it justify the method? What insight are we supposed to take away from that?

The experiments could be strengthened by addressing the following:
* What was epsilon during training? Why was epsilon = 0.05 in evaluation? This is quite high compared to previous work, and it makes sense that this would degrade MIRLs performance less than DQN and SQL.
* What is the performance of SQL if we use \rho as the action selector in \epsilon-greedy. This would help understand if the performance gains are due to the impact on the policy or due to the changes in the behavior policy.
* Plotting beta over time
* Comparing the action distributions for SQL and MIRL to understand the impact of the penalty. In general, a deeper analysis of the impact on the policy is important.
* Are their environments we would expect MIRL to outperform SQL based on your theoretical understanding? Does it?
* How many seeds were run per game?
* How and why were the 19 games selected from the full set?

Comments:

The abstract claims state-of-the-art performance, however, what is actually shown is that MIRL outperforms DQN and SQL.

With a fixed prior, the action prior can be absorbed into the reward (e.g., Levine 2018), so it is of no loss of generality to assume a uniform prior.

Could state that the stationary distribution is assumed to exist and be unique.

In Sec 3.1, why is the prior state independent?

In Sec 3.1, p(R = 1|\tau) is defined to be proportional to exp(\beta \sum_t r_t). Is this well-specified? How would we compute the normalizing constant since p(R = 0 | \tau) is not defined?

Throughout, I suggest that the authors not use the phrases "closed form" and "analytic" for expressions that are in terms of intractable quantities.

It should be noted that Sec 3.2 Optimal policy for a fixed prior \rho follows from Levine 2018 and others by transforming the fixed prior into a reward bonus.

In Sec 3.2, the last statement does not appear to be necessary for the next subsection. Remove or clarify?

I believe that the connection to MI can be simplified. Plugging in the optimal \rho into Eq 3, we can see that Eq 3 simplifies to \max_\pi E_q[ \sum_t \gamma^t r_t] - (1 - gamma)/\beta MI_p(s, a) where p(s, a) = d^\pi(s) * \pi(a | s) and d^\pi is the discounted state visitation distribution. Thus Eq 3 can be thought of as a lower bound on the MI regularized objective.

In Sec 4, the authors state the main difference between their soft operator and the typical soft operator. What other differences are there? Is that the only one?

Sec 5 references the wrong Haarnoja reference in the first paragraph.

In Sec 5, alpha_beta = 3 * 10^5. Is that correct?

=====
11/26
At this time, the authors have not responded to the reviews. I have read the other reviews and comments, and I'm not inclined to change my score.

====
12/7
The authors have addressed most of my concerns, so I have raised my score. I'm still concerned that the exploration epsilon is quite different than existing work (e.g., https://github.com/google/dopamine/tree/master/baselines).

---

> ### Author Response · Authors · 2018-11-26
> **Reply**
>
> We are sorry for the delayed reply (the deadline was extended to the end of 26th November Anywhere on Earth time). We state the reviewers comments and denote with arrows ( --------->  ) our replies.
>
> The authors take the control-as-inference viewpoint and learn a state-independent prior (which is typically held fixed). They claim that this leads to better exploration when actions have different importance. They relate this objective to a mutual information constrained RL objective in a limiting case. They then propose a practical algorithm, MIRL and compare their algorithm against DQN and Soft Q-learning (SQL) on 19 Atari games and demonstrate improvements over both.
>
> Generally I found the idea interesting and at a high level the deficiency of entropy regularization makes sense. However, I had great trouble understanding the reasoning behind their method and did not find the connection to mutual information helpful. Furthermore, I had a number of questions about the experiments. If the authors can clarify their motivation and reasoning and strengthen the experiments, I'd be happy to raise my score.
>
> In Sec 3.1, why is it sensible to optimize the prior? Can the authors give intuition for maximizing \log p(R = 1) wrt to the prior? This is critical for justifying their approach. Currently, the authors provide a connection to MI, but don't explain why this matters. Does it justify the method? What insight are we supposed to take away from that?
>
> -------------> [On prior optimization and mutual-information] We extended the paper with an explanation on mutual information and rate distortion theory, in order to help with an intuitive understanding of why this prior can help learning. We also added a related work section to note that other algorithms have considered optimizing the ELBO with respect to both variational and prior policy. However, these approaches do not use the marginal prior or have any connection to mutual information but instead optimise the policy while staying close to the previous policy. Additionally, we moved the connection to Mutual information for the case of gamma -> 1 to the appendix, and adopted another way to show this connection similar to what the reviewer has proposed.
>
>
>
> The experiments could be strengthened by addressing the following:
> * What was epsilon during training? Why was epsilon = 0.05 in evaluation? This is quite high compared to previous work, and it makes sense that this would degrade MIRLs performance less than DQN and SQL.
>
> ----------->[Epsilon in training and evaluation] Epsilon during training was decayed from 1.0 to 0.1 over the first 10^6 steps of the experiment. We used a fixed evaluation epsilon of 0.05. This procedure is standard in the literature for ATARI, as introduced by the DQN paper (see e.g. Mnih et al, 2015 ). We understand that in later DQN papers (e.g. Rainbow) different values for these hyperparameters have been used but we feel our choice is not unreasonable.
>
>
>
> * What is the performance of SQL if we use \rho as the action selector in \epsilon-greedy. This would help understand if the performance gains are due to the impact on the policy or due to the changes in the behavior policy.
>
> ----------->[On marginal exploration] We have run additional experiments combining SQL with marginal exploration. Using the marginal exploration helps SQL, but MIRL still achieves the best performance.
>
> * Plotting beta over time
> ----------->[Plotting beta] We include the beta values evolving over time in the appendix. Additionally, we also include a more relevant term (beta x Qvalues).
> * Comparing the action distributions for SQL and MIRL to understand the impact of the penalty. In general, a deeper analysis of the impact on the policy is important.
> * Are their environments we would expect MIRL to outperform SQL based on your theoretical understanding? Does it? * How many seeds were run per game?
> ----------->[Policy and grid world] Responding the previous two questions: We have added additional experiments and plots to the paper in an effort to  provide more insight into the behavior of our method. These experiments include a simple grid world in which we expect MIRL to outperform SQL and a grid world in which we expect the prior to have negative effects (as suggested by another reviewer).
> * How and why were the 19 games selected from the full set?
> ------------->[On other aspects] Due to computational constraints we were not able to run experiments on the full set of ATARI games. Therefore, we selected a subset of 20 random games, without prior experimentation on any of the games. We then evaluated our method using a single seed for every game. Data for experiments on 1 game were lost because of a cloud instance failure.

---

> > ### Author Response · Authors · 2018-11-26
> > **Continuation**
> >
> > Comments:
> > The abstract claims state-of-the-art performance, however, what is actually shown is that MIRL outperforms DQN and SQL.
> >
> > ---------->[Attenuated wording] We have adjusted the formulation regarding the performance in the paper.  We outperform DQN and SQL, both recent and high-performing algorithms (though not the best algorithms on ATARI). Our normalized scores are also close to those reported in the recent state-of-the art RAINBOW paper, but we cannot make a direct comparison over different implementations and subsets of games.
> >
> >
> >  With a fixed prior, the action prior can be absorbed into the reward (e.g., Levine 2018), so it is of no loss of generality to assume a uniform prior.
> >
> > --------------->[Absorbing prior into reward] In case of a uniform prior that is unaffected in the course of training, this is possible. In our algorithm, the prior is adapted in the course of training. In this case, keeping the prior separate allows for overcoming the problem of non-stationarity in the reward function.
> >
> > Could state that the stationary distribution is assumed to exist and be unique.
> >
> > ------------>[Unique stationary state distribution] We state now in the paper that the stationary distribution is assumed to exist and be unique.
> >
> >
> > In Sec 3.1, why is the prior state independent?
> > ---------->[State-independent prior] We base our formulation on the rate-distortion framework that generalizes entropy regularization by having optimal state independent priors. We provide some intuition for the one-step decision-making case in the background section.
> >
> >
> > In Sec 3.1, p(R = 1|\tau) is defined to be proportional to exp(\beta \sum_t r_t). Is this well-specified? How would we compute the normalizing constant since p(R = 0 | \tau) is not defined?
> >
> > ----------->[Normalization constant] It is not required to compute the normalization constant explicitly since it would appear in Equation 5 as a constant that is unaffected by the optimization. More explicitly, the expectation of the log of the normalization constant of p(R=1|\tau) w.r.t. q(\tau) is just the log of the normalization constant of p(R=1|\tau) without the expectation.
> >
> > Throughout, I suggest that the authors not use the phrases "closed form" and "analytic" for expressions that are in terms of intractable quantities.
> >
> > ----------->[Wording] We modified the wording accordingly in the current version of the  paper.
> >
> > It should be noted that Sec 3.2 Optimal policy for a fixed prior \rho follows from Levine 2018 and others by transforming the fixed prior into a reward bonus.
> >
> > In Sec 3.2, the last statement does not appear to be necessary for the next subsection. Remove or clarify?
> > ---------->[[Clarity] We added some clarifications to this section.
> >
> > I believe that the connection to MI can be simplified. Plugging in the optimal \rho into Eq 3, we can see that Eq 3 simplifies to \max_\pi E_q[ \sum_t \gamma^t r_t] - (1 - gamma)/\beta MI_p(s, a) where p(s, a) = d^\pi(s) * \pi(a | s) and d^\pi is the discounted state visitation distribution. Thus Eq 3 can be thought of as a lower bound on the MI regularized objective.
> > ----------->[On simplified connection to MI] We moved the connection to Mutual information for the case of gamma -> 1 to the appendix, and adopted another way to show this connection similar to what the reviewer has proposed.
> >
> > In Sec 4, the authors state the main difference between their soft operator and the typical soft operator. What other differences are there? Is that the only one?
> > ------------>The two main differences are an adaptive prior and adaptive beta.
> >
> > Sec 5 references the wrong Haarnoja reference in the first paragraph. In Sec 5, alpha_beta = 3 * 10^5. Is that correct?
> > ----------->We corrected this typo. It should be 3*10^-5.

---

> > > ### Comment · AnonReviewer1 · 2018-12-11
> > > **RE:**
> > >
> > > These answers address my questions.

---

> > ### Comment · AnonReviewer1 · 2018-12-11
> > **RE:**
> >
> > 1. Great, the changes have improved clarity.
> >
> > 2. The values are substantially different than previous work. See here for a summary of previous settings (https://github.com/google/dopamine/tree/master/baselines). This raises a red flag for the experiments.
> >
> > 3. Great, appreciate the new experiments.

---

> > > ### Author Response · Authors · 2018-12-12
> > > **reply**
> > >
> > > We thank the reviewer for appreciating the improvements of the paper.
> > >
> > >
> > > The attached link indeed shows a different epsilon value for evaluation (and other hyperparameters) used in this particular DQN implementation. An epsilon value for evaluation that differs from 0.05 was used in some of the previous literature (e.g. distributed DQN in Bellemare et al. 2017, prioritized double DQN in Schaul et al. 2016). However, earlier DQN papers do report an epsilon value of 0.05 for evaluation (original DQN in Mnih et al. 2015, double DQN in van Hasselt et al. 2016, prioritized DQN in Schaul et al. 2016). While an epsilon value of 0.01 might improve evaluation results, we feel a value of 0.05 is not unreasonable since we compare all methods under the same evaluation procedure. Additionally, we chose the other hyperparameters following the original DQN paper (Mnih et. al. 2015).

---

> ### Author Response · Authors · 2018-12-12
> **reply**
>
> We are thankful to the reviewer for noticing the improvements and raising the score.

---

### Official Review · AnonReviewer2 · 2018-11-02
**Simple approach that appears to work well**

**Rating:** 6
**Confidence:** 4

**Review:**

This work introduces SoftQ with a learned, state-independent prior. One derivation of this objective follows standard approaches from an RL as inference to derive the ELBO objective.

A more novel view derived here connects this objective with the rate-distortion problem to view the objective as an RL objective subject to a constraint on the mutual information between the state and action distribution.

They also outline a practical off-policy algorithm for optimizing this objective and compare it with Soft Q Learning (essentially, the same method but with a flat-prior) and DQN. They find that this results in small gains across most Atari games, with big gains for a few games.

This work is well-explained except in one-aspect. The rate-distortion view of the objective is not well-justified. In particular, why is it desirable in the context of RL to constrain this mutual information?

Empirical Deep RL performance is notoriously difficult to test (e.g. Henderson et al., 2017). The hyper-parameters are simply stated here, but no justification is given for how they are chosen / whether the baselines perform better under different choices. Given the gains compared with SoftQ are not that large, this information is important for understanding how much weight to place on the empirical result.

The fact that the prior does not converge in some environments (e.g. Seaquest) is noted, but it seems this bears further discussion.

Overall it appears this work provides:
- An algorithm for Soft Q learning with a learned independent prior
- Moderate evidence for gains compared with a flat prior on Atari.
- A connection with this approach and regularization by constraining the mutual information between state and action distributions.

It could be made a stronger piece of work by showing improvements in domains others than Atari, justifying the choice of regularization more. It would also benefit from positioning this work more clearly in relation to related approaches such as MPO (non-parametric state-dependent prior) and DistRL (state-dependent prior but shared across all games).

---

> ### Author Response · Authors · 2018-11-26
> **Added Grid World experiments, Related Work section and better connection to Mutual Information**
>
> We thank the reviewer for the comments.  Below we attempt to address each of the points raised by the reviewer.
>
> Background and related work:
>
> We have expanded the paper with a section highlighting the connection between the rate distortion framework and the mutual information constraint. We hope that this connection can help providing some intuitive insight into why our method can improve performance.
>
> We have also added a related work section more clearly positioning our work with respect to existing algorithms (such as MPO and DistRL).
>
> Experiments:
>
> We have included a new set of experiments on a small tabular domain.  While simple, we hope that this domain can provide more insight into the performance of the algorithm.
>
>
> Due to computational constraints we were not able to perform a complete search for optimal hyperparameter combinations in the Atari domain. Hyperparameter values were chosen by using values reported in the literature. Values for the new parameters introduced by MIRL were fixed by running a small number of exploratory experiments. Overall, we found the algorithm to be robust to changes in these values. All other hyperparameters were kept the same for all algorithms.
>
>
> While it is true that the prior does not converge in all of our ATARI experiments, we note that during the later stages of learning the plots do show a higher probability for subsets of actions. We have empirically observed that convergence of the prior can take a very long time, especially when the learner is still improving.  We expect that, given enough time, the probabilities of the marginal policy will eventually settle. Additionally, in these experiments we used a non-decaying learning rate for the marginal policy. This means that we can expect some oscillation due to tracking behaviour of our approximation, while the policy and state distribution still change.

---

> > ### Comment · AnonReviewer2 · 2018-12-10
> > **Response to author's**
> >
> > Thank you for response.
> >
> > I think this mostly addresses the concerns I raised.
> >
> > I appreciate the additional information regarding the rate-distortion, although I'm not sure that this view is adding much over the more usual view (why limit the rate of information encoded by policy?).
> >
> > Overall, I think this is interesting work and now better addresses prior work.
> >
> > My score was marginally positive, and I remain at this mostly due the idea being relatively straightforward and the gains being fairly marginal.

---

> > > ### Author Response · Authors · 2018-12-12
> > > **reply**
> > >
> > > We thank the reviewer for the feedback leading to improvements of the paper.
> > >
> > > In the final version, we will add a couple of additional sentences clarifying why a limit on information rate might be beneficial at initial stages of learning. In short, in prior work, it has been shown that the rate-distortion framework improves generalization in a supervised learning setting (Leibfried and Braun 2016). The intuition is that limits in transmission rate prevent overfitting on the training set. Similarly, in our work for the RL setting, limits in transmission rate prevent the agents to bootstrap with a ‘harsh’ max-operator that would lead to overestimation and sample inefficiency, but instead use a softened version less prone to overestimation with an adaptive prior that additionally improves exploration.

---

### Official Review · AnonReviewer3 · 2018-11-03
**Interesting idea, more experimental results needed**

**Rating:** 7
**Confidence:** 4

**Review:**

** Summary: **

The authors use the reformulation of RL as inference and propose to learn the prior policy. The novelty lies in learning a state-independent prior (instead of a state-dependent one) that can help exploration in the presence of universally unnecessary actions. They derive an equivalence to regularizing the mutual information between states and actions.

** Quality: **
The paper is mathematically detailed and correct.

** Clarity: **
The paper is sufficiently easy to follow and explains all the necessary background.

** Originality & Significance: **
The paper proposes a novel idea: Using a learned state-independent prior as opposed to using a learned state-dependent prior. While not a big change in terms of mathematical theory, this could lead to positive and interesting results empirically for exploration. Indeed they show promising results on Atari games: It is easy to see how Atari games could benefit as they have up to 18 different actions, many of which are redundant.

My two main points where I think the paper could improve are:
- More experimental results, in particular, how strong are the negative effects of MIRL if we have actions that are important, but have a lower probability in the stationary action distribution?
- A related work section comparing their approach to the many recent similar papers in Maximum Entropy RL

---

> ### Author Response · Authors · 2018-11-26
> **Added results on Grid World and added Related Work section**
>
> We thank the reviewer for the comments.
>
> We have updated the manuscript with additional experiments in a grid-world domain aimed at answering the reviewer’s concerns. The additional experiments are aimed at better understanding the behaviour of our mutual-information constraint. We demonstrate that our method clearly improves learning speed when there is a strong preference for a single action in the optimal policy.  We also examine an example in which the optimal policy crucially depends on an action with low probability in the marginal distribution. While MIRL does not improve performance in this case, it does not exhibit negative effects. We show that the learnt policy overcomes the prior when necessary for performance.
>
> Additionally, we have added a related work section that positions and compares our work to the existing literature on inference-based RL and maximum entropy RL in particular.

---

> > ### Comment · AnonReviewer3 · 2018-12-11
> > **Raised score to 7**
> >
> > I would like to thank the authors for their comments (both to mine and other's reviews) and the updated paper.
> >
> > The changes improve the paper, correspondingly I raised my score from 6 to 7.
> >
> > However, I still believe that more informative experiments about the limitations and drawbacks of the proposed method would highly increase the value to the community as it would allow readers to better judge whether the method should be incorporated in their work and, more importantly, it could point towards further research opportunities to improve on the presented work.
> > Consequently, I would strongly encourage the authors to incorporate such experiments in their CRC version if the paper gets accepts.
> > (I don't believe the current gridworld experiment actually shows the limitations as its reward structure doesn't discourage the infrequent action only until _after_ the first and only reward was already found).

---

> > > ### Author Response · Authors · 2018-12-12
> > > **reply**
> > >
> > > We thank the reviewer for raising the score and for the additional suggestions on analyzing potential limitations and drawbacks of our method. We will include a paragraph clarifying where our method might fail according to our pilot experiments, and perform additional experiments with a reward structure discouraging an infrequent action that is required to eventually succeed.

---

### Public Comment · (anonymous) · 2018-10-23
**Connection to prior work**

Hello,

Thanks for the paper. I would like to point out a paper from  ICLR2018 that shares similarities in both

1- The derivations of RL objective from Inference perspective
2- The resulting objective function for learning the prior

please see,

Maximum a-Posteriori Policy Optimisaiton
https://arxiv.org/pdf/1806.06920.pdf

In the paper above, the mutual information (Or expected KL ) regularized objective is derived in E-step (see equation 7). And the optimal solution is given in (8) when a non parametric variational distribution is used.

It would be useful if authors discuss the connections and differences.

Thank you,

---

> ### Author Response · Authors · 2018-10-24
> **Differences between MPO and our approach**
>
> Thank you for your comment.
>
> Framing RL as an inference problem has been addressed before in the literature [1,2] and can be done in different ways.  The difference between the variational inference formulation in MPO and our variational inference formulation is the following:
> - The policy of the generative model in our case is state-independent (similar to [1]) with the optimal solution being the marginal distribution over actions ([1] does not consider an optimal marginal distribution though). In contrast, in MPO the generative policy is state-dependent and given by the previous-round behavioural policy.
>
> Importantly, our specific choice of state-dependent variational policy and state-independent generative policy directly leads to a mutual information regularizer. Note that the mutual information is not any expected KL, but a specific expected KL under the assumption of an optimal marginal policy (which is exactly what we model). MPO does not have the notion of an optimal marginal policy (in the sense of a state-independent marginal policy) and therefore the expected KL in MPO is not a mutual information.
>
> In our experimental section we empirically validate that our mutual information regularized objective leads to improvements over soft-q learning (see [1]) where the generative policy is also state-independent but not subject to optimization (but instead given by a uniform distribution).
>
> We will clarify this point in a revised version of the manuscript.
>
> [1] Levine, S. Reinforcement Learning and Control as Probabilistic Inference: Tutorial and Review. arXiv 2018.
> [2] Neumann, G. Variational Inference for Policy Search in changing Situations. ICML 2011.

---

> > ### Public Comment · (anonymous) · 2018-10-24
> > **Question**
> >
> > Thank you very much for the reply.
> >
> > Then if MPO use a state-independent generative policy, it will reduce to the proposed algorithm?
> > I understand that a learned state-independent generative policy is better than a uniform one. My question is that, why state-independent generative policy should be better than state-dependent generative policy as used by MPO?

---

> > > ### Author Response · Authors · 2018-10-29
> > > **Reply**
> > >
> > > Both our algorithm and MPO can be seen as optimizing the same  evidence lower bound (ELBO).  MPO proposes a general coordinate ascent type optimization in which the ELBO is updated in alternating steps, either with respect to the variational policy or the prior policy (while the other policy is kept fixed). Different design choices for the policies and optimization procedures give rise to different, but related algorithms. This approach is also common in variational inference based policy search and describes a large family of related policy search algorithms (see Deisendroth et al, 2013 for an overview.)
> > >
> > > Our algorithm follows recent soft Q-learning algorithms (e.g. Fox et al, 2016, Haarnoja et al. 2017). These algorithms consider the same ELBO, but omit the optimization with respect to the prior policy and only optimize the variational policy pi.  This can be seen as an entropy-regularized version of standard Q-learning algorithms.  When the prior is fixed to be a constant uninformative policy, this procedure reduces to max-entropy policy learning. The algorithm replaces the classic Bellman operator with a soft Bellman-operator to prevent deviations from a state-independent fixed prior policy. Several papers (e.g.Haarnoja et al 2017, Schulman et al 2017 ) have shown that these “softened” algorithms offer advantages over their unsoftened counterparts, in terms of exploration, generalization and composability. Our approach further improves on soft Q-learning (as shown in our Atari experiments) by allowing for optimizing the prior (while still being state-independent). As shown in the paper, this results in a mutual information constraint (rather than a max entropy constraint) on the resulting policy.
> > >
> > > So while we follow the same general scheme as soft Q-learning, we do update our prior policy as in the MPO algorithm. However, contrary to MPO, we do not consider the alternating, coordinate descent style optimization. Rather than executing a separate prior maximization step, we solve the ELBO for the optimal prior in the special case of state-independent priors.  We then directly estimate this optimal prior in our algorithm, instead of performing a gradient style update on the ELBO. While it is possible to consider the same class of state-independent priors with MPO, the way in which both algorithms optimize the ELBO will still be different.
> > >
> > > A modified MPO that uses a state-independent generative policy would converge to a solution that is penalized by an optimal marginal policy. However, since the parameter epsilon (that determines the deviation between the variational and the generative policy) is fixed and not scheduled in the course of training, the final solution is still constrained by the marginal policy which is sub-optimal because it is state-independent. This constraint would essentially limit the asymptotic performance of such a modified MPO. Of course, this could be alleviated by setting epsilon to a large value but this would correspond to an ordinary actor critic  approach without any regularization in the policy.
> > >
> > > If the prior policy in our algorithm is replaced by a state-dependent prior, the optimal solution for such a prior is the variational policy (i.e. pi) itself. This essentially would eliminate the KL-constraint and reduce our algorithm to standard Q-learning. Q-learning is known to suffer from sample-inefficiency caused by the hard max-operator in the target (this leads to overestimated q-values). This is exactly the problem that was been addressed by soft Q-learning with entropy regularization.
> > >
> > > Deisenroth, M. P., Neumann, G., & Peters, J. (2013). A survey on policy search for robotics. Foundations and Trends® in Robotics, 2(1–2), 1-142.
> > >
> > > Schulman, J., Chen, X., & Abbeel, P. (2017). Equivalence between policy gradients and soft q-learning. arXiv preprint arXiv:1704.06440.
> > >
> > > Haarnoja, T., Tang, H., Abbeel, P., & Levine, S. (2017). Reinforcement learning with deep energy-based policies. arXiv preprint arXiv:1702.08165.
> > >
> > > Fox, Roy, Ari Pakman, and Naftali Tishby. Taming the noise in reinforcement learning via soft updates. UAI (2016).

---

### Meta-Review · Area_Chair1 · 2018-12-14
**Interesting contribution that improves on the widely used entropy regularized algorithms**

**Confidence:** 4
**Recommendation:** Accept (Poster)

**Metareview:**

The paper proposes a new RL algorithm (MIRL) in the control-as-inference framework that learns a state-independent action prior.  A connection is provided to mutual information regularization.  Compared to entropic regularization, this approach is expected to work better when actions have significantly different importance.    The algorithm is shown to beat baselines in 11 out of 19 Atari games.

The paper is well written.  The derivation is novel, and the resulting algorithm is interesting and has good empirical results.  A few concerns were raised in initial reviews, including certain questions about experiments and potential negative impacts of the use of nonuniform action priors in MIRL.  The author responses and the new version were quite helpful, and all reviewers agree the paper is an interesting contribution.

In a revised version, the authors are encouraged to
  (1) include a discussion of when MIRL might fail, and
  (2) improve the related work section to compare the proposed method to other entropy regularized RL (sometimes under a different name in the literature), for example the following recent works and the references therein:
    https://arxiv.org/abs/1705.07798
    http://proceedings.mlr.press/v70/asadi17a.html
    http://papers.nips.cc/paper/6870-bridging-the-gap-between-value-and-policy-based-reinforcement-learning
    http://proceedings.mlr.press/v80/dai18c.html